# In silico analysis suggests less effective MHC-II presentation of SARS-CoV-2 RBM peptides: Implication for neutralizing antibody responses

Andrea Castro [1,2], Kivilcim Ozturk[2], Maurizio Zanetti[3,4]*, Hannah Carter[2,4]*

1 Biomedical Informatics Program, University of California San Diego, La Jolla, CA, United States of America, 2 Division of Medical Genetics, Department of Medicine, University of California San Diego, La Jolla, CA, United States of America, 3 The Laboratory of Immunology, Department of Medicine, University of California San Diego, La Jolla, CA, United States of America, 4 Moores Cancer Center, University of California San Diego, La Jolla, CA, United States of America

* mzanetti@health.ucsd.edu (MZ); hkcarter@health.ucsd.edu (HC)

**Data Availability Statement:** All relevant data are within the manuscript and its Supporting Information files.

## Abstract

SARS-CoV-2 antibodies develop within two weeks of infection, but wane relatively rapidly post-infection, raising concerns about whether antibody responses will provide protection upon re-exposure. Here we revisit T-B cooperation as a prerequisite for effective and durable neutralizing antibody responses centered on a mutationally constrained RBM B cell epitope. T-B cooperation requires co-processing of B and T cell epitopes by the same B cell and is subject to MHC-II restriction. We evaluated MHC-II constraints relevant to the neutralizing antibody response to a mutationally-constrained B cell epitope in the receptor binding motif (RBM) of the spike protein. Examining common MHC-II alleles, we found that peptides surrounding this key B cell epitope are predicted to bind poorly, suggesting a lack MHC-II support in T-B cooperation, impacting generation of high-potency neutralizing antibodies in the general population. Additionally, we found that multiple microbial peptides had potential for RBM cross-reactivity, supporting previous exposures as a possible source of T cell memory.

## Introduction

Upon infection with SARS-CoV-2 the individual undergoes seroconversion. In mildly symptomatic patients, seroconversion occurs between day 7 and 14, includes IgM and IgG, and outlasts virus detection with generally higher IgG levels in symptomatic than asymptomatic groups in the early convalescent phase [1]. Alarmingly, the IgG levels in both asymptomatic and symptomatic patients decline during the early convalescent phase, with a median decrease of ~75% within 2–3 months after infection [2]. This suggests that the systemic antibody response which follows natural infection with SARS-CoV-2 is short-lived, with the possibility of no residual immunity after 6–12 months [3] affecting primarily neutralizing antibodies in

**Funding:** This work was supported by an NIH National Library of Medicine Training Grant T15LM011271 to A.C., an Emerging Leader Award #18-022-ELA from The Mark Foundation for Cancer Research (https://themarkfoundation.org), a Canadian Institute For Advanced Research (CIFAR) (https://www.cifar.ca) fellowship #FL-000655 to H.C. and NIH NCI RO1 CA220009 to M. Z. and H.C.

**Competing interests:** The authors have declared that no competing interests exist.

plasma [4]. Early activated B cells produce antibodies in quasi-germline configuration and are likely 'innate-like B cells' [5–8] that have not undergone somatic hypermutation and maturation. Consistent with the above argument, a lack of germinal center formation but robust activation of non-germinal type B cells has been reported in cases of severe COVID-19 infection, impairing production of long-lived memory or high affinity B cells [9].

The generation of an antibody response requires cooperation between a B cell producing specific antibody molecules and a CD4 T cell (helper cell) activated by an epitope on the same antigen as that recognized by the B cell (T-B cooperation) [10]. This reaction occurs in the germinal center [11, 12]. Excluded from this rule are responses against carbohydrates and antigens with repeating motifs that alone cross-link the B cell antigen receptor leading to B cell activation [13]. Discovered over 50 years ago [14–16], it also became apparent that T-B cooperation is restricted by Major Histocompatibility Complex class II (MHC-II) molecules [17–19]. T-B cooperation plays a key role in the facilitation and strength of the antibody response [15, 20] and the size of the antibody response is proportional to the number of Th cells activated by the B cell during T-B cooperation [18, 19, 21]. The importance of T cell help during the activation of antigen specific B cells to protein antigens driving B cell selection is emphasized by recent experiments where the injection of a conjugate of antigen (OVA) linked with an anti-DEC205 antibody induced a greater proliferation of DEC205+ relative to DEC205- B cells consistent with a T helper effect on B cell activation [22].

T-B cooperation requires that the epitopes recognized by the B and T cell be on the same portion of the antigen [16, 23, 24] leading to a model requiring the contextual internalization and co-processing of T and B cell epitopes [10] which is consistent with the principle of linked (aka associative) recognition of antigen [25]. Studies *in vitro* using human T and B lymphocytes showed that an antigen specific B cell can present antigen to CD4 T cells even if antigen is present at very low concentration ($10^{-11}$–$10^{-12}$ M) [26]. Presentation of antigen by the B cell also facilitates the cooperation between CD4 T cells of different specificities resulting in enhanced generation of memory CD4 T cells [27]. However, T-B cooperation is not the only form of cooperative interaction among lymphocytes as cooperation exists between CD4 T and CD8 T cells [28] and between two CD4 T cells responding to distinct epitopes on the same antigen [29].

A model based on coprocessing of T and B epitopes also led to the suggestion that preferential T-B pairing could be based on topological proximity [30–34] so that during BCR-mediated internalization the T cell epitope is protected by the paratope of the BCR. Indeed, a more recent study showed that not only is CD4 T cell help a limiting factor in the development of antibodies to smallpox (vaccinia virus), but that there also exists a deterministic epitope linkage of specificities in T-B cooperation against this viral pathogen [35]. Collectively, it appears that T-B pairing and MHC-II restriction are key events in the selection of the antibody response to pathogens and that operationally T-B cooperation and MHC-II restriction are key events in the generation of an adaptive antibody response, suggesting that lack of or defective T-B preferential pairing could result in an antibody response that is suboptimal, short-lived, or both.

The relevance of T-B cooperation in protective antiviral responses has been documented in numerous systems. In the influenza A virus (PR8) system it was shown that while Th1 CD4 T cell responses on their own are ineffective at promoting recovery from infection, antibodies generated through T-B cooperation were indispensable in the protective response against the virus [36]. In a different influenza A strain, it was shown that T-B cooperation and CD4 T cells represent a limiting factor in the kinetics and early magnitude of the primary B cell response to virus challenge and provide help in a preferential way (i.e. intra-molecular but nor inter-molecular) [37]. Additionally, CD40-CD40L (costimulatory molecules found on B cells and

CD4 T cells, respectively) interaction is required for the generation of antibody responses, isotype switching and memory responses in non-viral model systems [38]. In LCMV (lymphocytic choriomeningitis virus) and VSV (vesicular stomatitis virus) abrogation of CD40-CD40L interaction prevented T-B cooperation and thus inhibited antiviral protection [39]. Interestingly, this study also showed that the activation of CD4 T cells (e.g., inflammatory CD4 T cells) not associated with the activation of B cells was not compromised [39]. These data demonstrate the relevance of T-B cooperation in the antibody response in protection against viral infection.

In SARS-CoV-2, neutralizing antibodies (NAbs) are a key defense mechanism against infection and transmission. NAbs generated by single memory B cell VH/VL cloning from convalescent COVID-19 patients have been extremely useful in defining the fine epitope specificity of the antibody response in COVID-19 individuals. At present, SARS-CoV-2 NAbs can be distinguished into three large categories. 1) Repurposed antibodies, that is, NAbs discovered and characterized in the context of SARS-CoV and subsequently found to neutralize SARS--CoV-2 via cross-reactivity. These antibodies map away from the receptor binding domain (RBD) of the spike protein [40–42]. 2) Non-RBD neutralizing antibodies discovered in SARS--CoV-2 patients whose paratope is specific for sites outside the RBD [43]. 3) RBD antibodies, including NAbs, derived from SARS-CoV-2 patients that map to a restricted site in the RBD [7, 44–49]. Cryo-EM of this third antibody category shows that they bind to residues in or around the four amino acids Phe-Asp-Cys-Tyr (FNCY) in the receptor binding motif (RBM) (residues 437–508) which is inside the larger RBD (residues 319–541) at the virus:ACE2 interface [45]. Although the RBD has been shown to be an immunodominant target of serum antibodies in COVID-19 patients [50], high potency NAbs are directed against a conserved portion of the RBM on or around the FNCY patch, a sequence only found in the RBD of SARS-CoV-2 and not in other coronaviruses. NAbs that make contact with the FNCY patch outperform other NAbs that do not in competition binding assays, highlighting the importance of the region in neutralizing ACE2 binding [43]. Indeed while the RBD is mutationally tolerant, the RBM is constrained to the wild-type amino acids [51], implying that the B cell epitope included in this region of the virus:ACE2 interface is resistant to antigenic drift. Thus, we may refer to this site as a key RBM B cell epitope in the generation of potent NAbs.

Antibody responses against SARS-CoV-2 depend on CD4 T cell help. Spike-specific CD4 T cell responses have been found to correlate with the magnitude of the anti-RBD IgG response whereas non-spike CD4 T cell responses do not [52]. However, in unexposed patients, spike-specific CD4 T cells reactive with MHC-II peptides proximal to the central B cell epitope represent a minority (~10%) of the total CD4 T cell responses, which are dominated by responses against either the distal portion of the spike protein or other structural antigens [53]. Surprisingly, these CD4 T cell responses are largely cross-reactive and originate from previous coronavirus infections [54].

As mounting evidence suggests that the NAb response in COVID-19 patients is relatively short-lived, we decided to test the hypothesis that associative recognition of a key RBM B cell epitope (in and around the FNCY patch) and proximal MHC-II-restricted epitopes may be defective with detrimental effects on preferential T-B pairing. Specifically, we hypothesize that the inability to present SARS-CoV-2 peptide sequences near putative B cell epitopes may impair memory cell generation and consequently reduce the strength and longevity of overall and neutralizing antibody responses. To quantify the potential effects of T-B cooperation *in vivo*, we analyzed all 15mer putative MHC-II epitopes (+/- 50 amino acid residues) relative to the key RBM B cell epitope for coverage by all known 5,620 human MHC-II alleles and predicted binding affinity. The analysis shows that there exists in general less availability of effective T cell epitopes in close proximity to the key RBM B cell epitope in the human population.

## Results

### Topology of a key RBM B cell epitope

Within the 222 amino acid long RBD of the spike protein (residues 319–541), the RBM (residues 437–508) is the portion of the spike protein that establishes contact with the ACE2 receptor (Fig 1A). The contact residues span a relatively large surface involving approximately 17 residues [45], among them residues F486, N487, Y489 form a loop, which we term the FNCY patch, which is surface exposed and protrudes up towards the ACE2 receptor from the bulge of the RBD (Fig 1B and 1C). F486 forms hydrophobic interactions with three ACE2 residues (L79, M82, W83). N487 forms hydrogen bonds with Q24 and W83, and Y489 is linked with K31 via a hydrophobic interaction. This makes the amino acid residues in or around the FNCY patch a logical B cell epitope target for antibodies blocking the virus:receptor interaction. In addition, these core residues are mutationally constrained by the ACE2 contact surface [51]. Not surprisingly, a set of recently reported potently

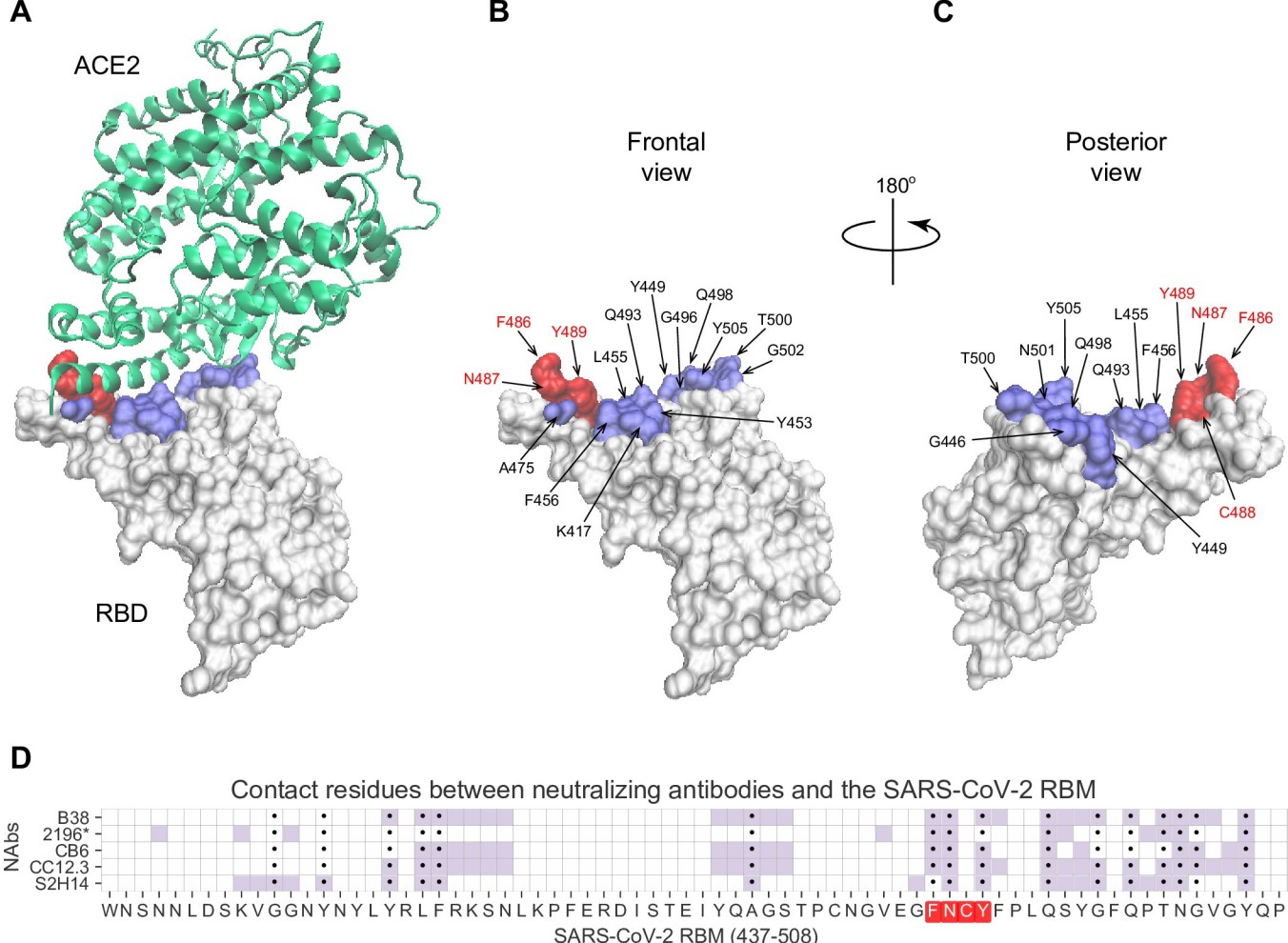

**Fig 1. Visualization of the FNCY core of the RBM B cell epitope on the SARS-CoV-2 spike protein RBD.** (A) 3D structure of the SARS-CoV-2 spike protein RBD (white) binding the ACE2 receptor (green) (PDB: 6M0J) with contact residues highlighted in blue and the FNCY patch highlighted in red. (B-C) Spike protein RBD with ACE2 contact residues and FNCY patch residues labeled in two orientations (front and back). (D) Heatmap of neutralizing antibody contact residues (purple) on the spike protein RBM region (positions 437–508). Black dots indicate ACE2 contact residues and the FNCY patch is highlighted in red. Source data available in S1 Table.

neutralizing antibodies generated by single B cell VH/VL cloning from convalescent COVID-19 patients all bear paratopes that include the FNCY patch in their recognition site [43, 47–49, 55] (**Fig 1D**). While other residues (Q493, N501, and Y505) are also shared between ACE2 and the paratope of these antibodies, they are not as protruding and are on a β-sheet unlike the FNCY patch which is organized in a short loop as a result of the C480: C488 disulfide bond. Thus, blockade of the RBM:ACE2 interaction (neutralization) depends at least in part on a B cell epitope in the RBM that is structurally and functionally critical to the interaction, virus internalization, and cell infectivity.

## Prediction of MHC-II affinity for 15mer peptides proximal to the RBM B cell epitope

In the T-B cooperation model, B cell activation and production of NAbs is dependent on CD4 T cell responses to MHC-II restricted peptides. To test the hypothesis that the generation of NAbs against a mutationally constrained B cell epitope in the RBM reflects the efficiency of processing and presentation of MHC-II peptides proximal to the FNCY patch, we evaluated the landscape of MHC-II peptide restriction across the entire SARS-CoV-2 spike protein with respect to common MHC-II alleles in the human population. To assess the potential for effective restriction by MHC-II molecules in a reasonable proportion of the population, we devised a position-based score that assigns each amino acid residue the median affinity of the best overlapping peptide, where median affinity is calculated across the 1911 most common MHC-II alleles (**Fig 2A**), which was highly correlated with scores across all 5620 MHC-II alleles (**Fig 2B**; Pearson rho = 0.99, p<2.2e-308). While a number of sites along the spike protein are predicted to generate high affinity peptides for most common MHC-II alleles, the region around the FNCY patch was depleted for generally effective binders (**Fig 2C**, Fisher's exact OR = 0.21, p = 0.015, Methods, **S1 Fig**). Interestingly, the RBM region containing the FNCY patch was free of glycans that could potentially mask the epitope (**Fig 2D**). We further evaluated the distributions of binding affinities for the 20 best-ranked peptides across all sites in the spike protein (**Fig 2E**), and in comparison, the distributions for the best 20 peptides overlapping positions within +/- 50 residues of the FNCY patch (**Fig 2F**). In the best case, less than half of the considered MHC-II alleles bound a shared peptide close to the FNCY patch, whereas at other sites there were multiple peptides that could be bound by nearly all of the MHC-II alleles (**Fig 2E**). This suggested overall less availability of effective T cell epitopes in close proximity to the FNCY B cell epitope, which could limit the availability of T cell help during an epitope-specific T-B cooperative interaction in the germinal center.

To further assess whether population variation in MHC-II MHC alleles might contribute to heterogeneity in potential to generate neutralizing antibodies, we also evaluated the potential of MHC-II supertypes to restrict peptides from neighboring the FNCY patch. Greenbaum *et al*. previously defined 7 supertypes that group MHC-II alleles based on shared binding repertoire. These 7 supertypes account for between 46%-77% of haplotypes and cover over 98% of individuals when all four loci are considered together [58]. We revisited our analysis of peptide restriction proximal to the FNCY patch treating each supertype separately. There was considerable variability in potential to effectively present FNCY patch proximal sequences across supertypes (**Fig 3A and 3B**, $X^2$ = 175, p = 3.75e-35, **S2 Fig**). Only 3 supertypes (DP2, main DP and DR4) commonly presented peptides overlapping the FNCY patch (**Fig 3B**). We were able to obtain population allele frequencies for four populations from the Be The Match registry [59] and Du *et al*. [60]. These data show that DR4 is relatively infrequent across the populations evaluated, whereas main DR, main DP, and DP2 are more common (**Fig 3C**), and thus could be more important for MHC-II restriction supportive of neutralizing antibodies. While

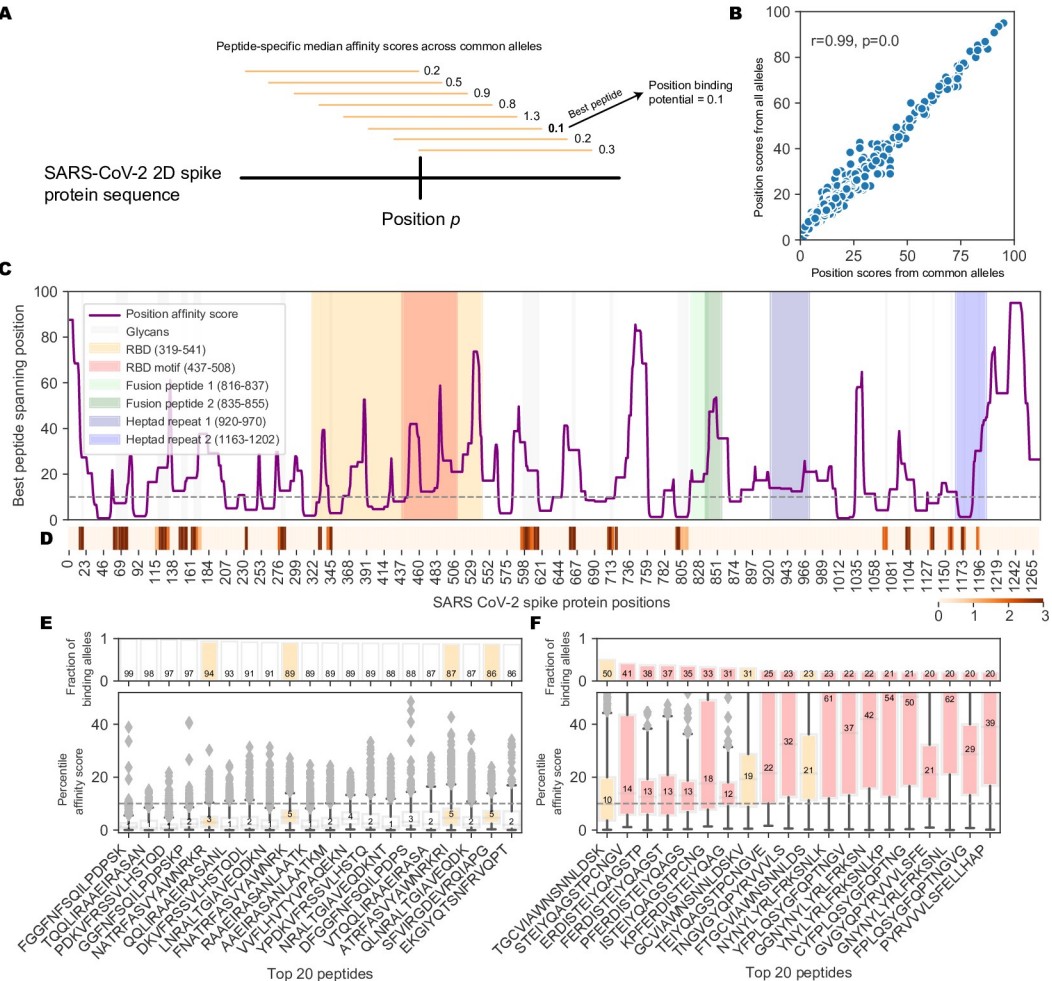

**Fig 2. Landscape of MHC-II binding affinity across spike protein 2D sequence.** (A) Overview of the position affinity score. (B) Scatterplot showing position affinity scores estimated using only common (>10% frequency, from [56]) MHC-II alleles (x-axis) versus across all MHC-II alleles (y-axis). (C) Lineplot showing the position affinity scores across common MHC-II alleles (Methods). Annotated domains from UniProt are highlighted. (D) Heatmap showing amino acid positions that are glycosylated [57]. (E) Barplots (top) and boxplots (bottom) describing the fraction of binding MHC-II alleles and corresponding affinity percentile rank distributions respectively for the top 20 peptides with the highest fraction of common binding alleles. The binding threshold of 10 is shown as a dotted line, with values less than 10 indicating binding. Colors correspond to the regions listed in C. (F) Barplots (top) and boxplots (bottom) describing the fraction of binding MHC-II alleles and corresponding affinity percentile rank distributions respectively for the top 20 peptides within +/-50 amino acids of the FNCY B cell epitope. Colors correspond to the regions listed in C.

there were some large population-specific differences in main DP and DP2 supertype frequencies, these frequency estimates are based on a limited population sample and may provide only a rough approximation. In general, DP and DR haplotypes were able to restrict more FNCY patch proximal sequences (**Fig 3D**).

## Cross-reactivity to a non-coronavirus MHC-II binding peptide as a potential driver of T cell responses helping antibody response to the RBM B cell epitope

Interestingly, Mateus *et al.* reported pre-existing CD4 T cell responses to peptides derived from the spike protein using T cells from unexposed individuals, suggesting previous

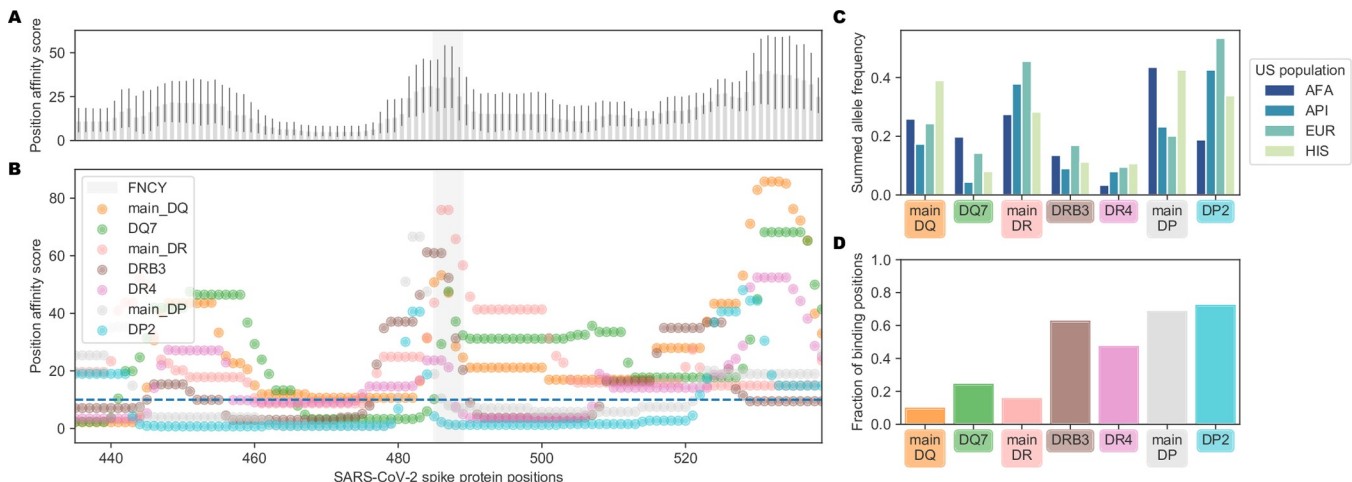

**Fig 3. Population variation affecting availability of FNCY proximal T cell epitopes.** (A) Barplot showing the aggregated supertype position affinity scores for each position +/- 50 amino acids from the FNCY patch (grey zone). (B) Scatterplot showing the specific supertype position scores for each position +/- 50 amino acids from the FNCY patch (grey zone). The binding threshold of 10 is shown as a dashed blue line, with points below the threshold indicating binding. (C) Barplot showing United States population frequencies, summed across the available alleles in each supertype. (D) Fraction of positions falling below the binding threshold within the region of interest for each supertype.

exposures to other human coronaviruses could potentially generate protective immunity toward SARS-CoV-2. Indeed, regions of higher coronavirus homology were associated with more T cell responses in their data [54]. This represents the most comprehensive interrogation of the spike protein with response to CD4 T cell responses to date. They screened all 15mers of the spike protein in pooled format and further evaluated 66 predicted MHC-II peptides that generated CD4 T cell responses. Visualizing the landscape of the CD4 T cell responses described in their work by percent positive response (**Fig 4A**) or spot forming cells (**Fig 4B**), we noted relatively few responses proximal to the FNCY patch in the RBM. Accordingly, few other coronaviruses had limited homology to the FNCY region, and none fully included the FNCY patch (**Fig 5A**).

A notable exception in Mateus' results is peptide 486FNCYFPLQSYGFQPT500, which was reported to induce a CD4 T cell response in an unexposed individual. In this case, the peptide

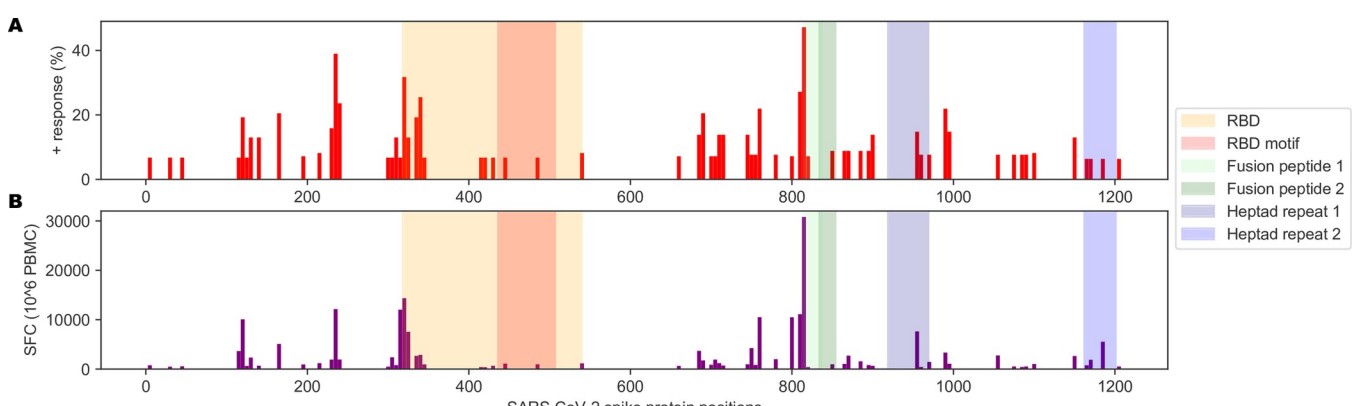

**Fig 4. Immunological history of relevance to SARS-CoV-2.** (A) Barplot showing the percentage of positive responses toward SARS-CoV-2 peptides from unexposed individuals. (B) Barplot showing the number of spot-forming cells (SFC) for tested SARS-CoV-2 peptides against PBMCs from unexposed individuals. Data from S1 Table from [54].

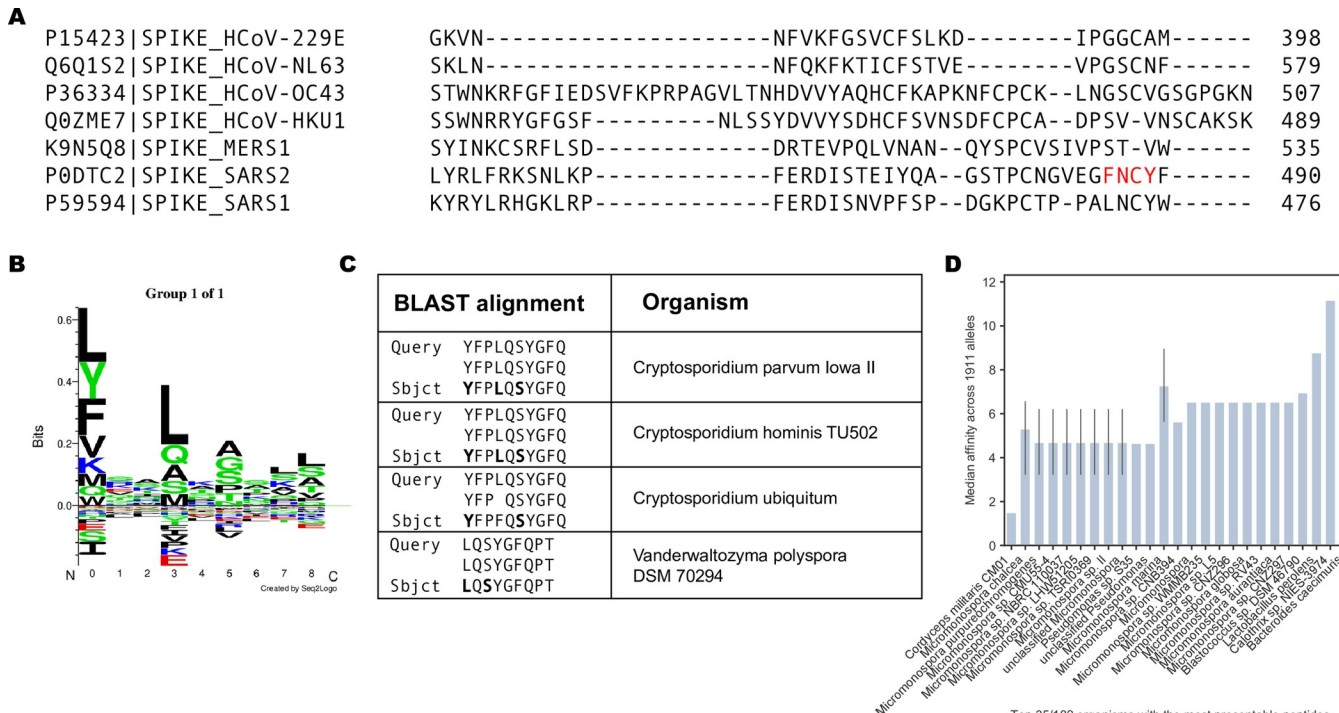

**Fig 5. Learned immunity to other targets that could support T cell responses to SARS-CoV-2.** (A) Multiple sequence alignment between SARS-CoV-2, SARS1, MERS, and other human coronaviruses, focusing on the region surrounding the FNCY B cell epitope. (B) SeqLogo plot obtained by clustering IEDB peptides reported to bind to DRB1*01:01. (C) Top results after blasting the FNCYFPLQSYGFQPT peptide against all reference proteins. (D) Barplot describing best peptide affinities across MHC-II alleles of the top 35 unique organisms with one or more peptides matching a peptide with high similarity to 15mers +/-30aa from the FNCY binding epitope based on BLAST analysis. The closer to 0, the greater the binding potential.

was restricted by HLA-DRB1*0101 or HLA-DQA1*0101/DQB1*0501. We found that the peptide sequence had greater *in silico* predicted affinity to HLA-DRB1*0101. To explain the conundrum, we blasted this peptide against the "refseq_protein" database excluding SARS-CoV-2 (Methods). Surprisingly, the sequences with the best homology for this query were not from coronaviruses but rather from common pathogens, first among them parasites of the *Cryptosporidium* genus of apicomplexan parasitic alveolates. These sequences included conserved anchor positions for the HLA-DRB*0101 allele making it plausible that a prior exposure could account for the formation of a memory CD4 T cell response (**Fig 5B and 5C**). To further assess the potential for other prior exposures in generating immune memory for sequences proximal to the FNCY patch we blasted all 15mers within +/-30 amino acids of the FNCY patch and filtered the resulting sequences based on restriction by consensus MHC-II supertypes [58] (**S2 Table**). We found peptides associated with multiple microbial organisms that may meet the criteria to potentially generate CD4 T cell memory relevant to the RBM of SARS-CoV-2 (**Fig 5D**).

## Discussion

SARS-CoV-2 uses the RBD of the spike protein to bind to the ACE2 receptor on target cells. The actual contact with ACE2 is mediated by a discrete number of amino acids that have been visualized by cryo-EM (Lan et al., 2020; Shang et al., 2020). Although several SARS-related coronaviruses share 75% homology and interact with ACE2 on target cells (Ge et al., 2013; Ren et al., 2008; Yang et al., 2015) the RBM in SARS-CoV-2 is unique to this virus. *In vitro* binding

measurements show that SARS-CoV-2 RBD binds to ACE2 with an affinity in the low nano-molar range (Walls et al., 2020). Mutations in this motif could be detrimental to the virus's ability to infect ACE2 positive human cells. Since the RBD is an immunodominant site in the antibody response in humans [50] it is not surprising that the paratope of some antibodies isolated from convalescent individuals via single B cell VH/VL cloning, and selected on the basis of high neutralization potency, all seem to bind a surface encompassing the FNCY patch in the RBM [7, 8, 44, 46–49]. Arguably, this motif corresponds to a relevant B cell epitope in the spike protein of SARS-CoV-2 and is a logical target of potent neutralizing antibodies.

Although antibodies directed to this site have been isolated by different groups, little is known about their contribution to the pool of antibodies in serum of SARS-CoV-2 infected individuals, but evidence suggests they are likely to be rare. In one study they were found to represent a subdominant fraction of the anti-RBD response [49] while the estimated frequency of antigen-specific B cells ranges from 0.07 to 0.005% of all the total B cells in COVID-19 convalescent individuals [61]. In a second study, the identification of two ultra-potent NAbs having a paratope involving the FNCY patch required screening of 800 clones from twelve individuals [8]. This suggests that a potent NAb response to a mutationally constrained RBM epitope is a rare component of the total anti-virus response consistent, with the observation that there is no correlation between RBM site-specific neutralizing antibodies and serum half-maximal neutralization titer (NT50) [61]. Here we show that the core RBM B cell epitope is apparently uncoupled from preferential T-B pairing, a prerequisite for a coordinated activation of B cells against the pathogen. We analyzed MHC-II binding of 15mer peptides in the spike protein upstream (-50 aa) or downstream (+50 aa) of the central RBM B cell epitope and found both low coverage by 1911 common MHC-II alleles and a depletion of binding 15mers proximal to the FNCY patch versus other exposed areas on the spike protein. This could be due to the fact that a sizeable proportion (40%) of CD4 T cells responding to the spike protein are memory responses found in SARS-CoV-2 unexposed individuals [52, 62] or other structural protein of SARS-CoV-2 such as the N protein [53]. Thus, it is possible that these conserved responses are used as a decoy mechanism to polarize the response away from the RBM. However, this does not rule out the contribution of a bias in frequency of specific B cells in the available repertoire.

Corroboration to our hypothesis also comes from Mateus *et al*. [54] who tested sixty-six 15mer peptides of the spike protein in SARS-CoV-2 unexposed individuals and found that CD4 T cell responses against this narrow RBM site account for only 2/110 (1.8%) of the total CD4 T cell response to 15mer peptides of the spike protein. Surprisingly, a CD4 T cell response against peptide FNCYFPLQSYGFQPT was by CD4 T cells of an unexposed individual. Since this peptide has low homology with previous human coronaviruses, we reasoned that this could either represent a case of TCR cross-reactivity since a single TCR can engage large numbers of unique MHC/peptide combinations without requiring degeneracy in their recognition [63, 64]. Remarkably, however, a BLAST analysis revealed a 10 amino acid sequence match with proteins from pathogens including those from the *Cryptosporidium* genus, with identity in binding motif and anchor residues (agretope) for the restricting MHC-II allele strongly suggesting peptide cross-reactivity. *Cryptosporidium hominis* is a parasite that causes watery diarrhea that can last up to 3 weeks in immunocompetent patients [65]. Additional possibilities for cross-reactivity to the RBM, albeit of a lesser stringency, involve antigens from *Micromonospora*, *Pseudomonas*, *Blastococcus*, *Lactobacillus, and Bacteroides* (**Fig 5D**). Thus, it appears as if memory CD4 T cells reactive with peptides in the RBM may reflect the immunological history of the individual that, as evidenced by this case, can be unrelated to infection by other coronaviruses. Interestingly, the great majority (64–88%) of COVID-19 positive individuals in homeless shelters in Los Angeles and Boston were found to be asymptomatic [66]. This suggests that

the status of the immune system, which itself reflects past antigenic exposure, may be a determining factor in the generation of a protective immune response after SARS-CoV-2 infection.

The findings reported herein have considerable implications for natural immunity to SARS-CoV-2. The fact that there seems to be an overall suboptimal T-B preferential pairing suggests that B cells that respond to the RBM B cell epitope may receive inadequate T cell help. This is consistent with the observation that in general potent neutralizing antibodies to the RBM undergo very limited somatic mutation [8, 46] and are by and large in quasi-germline configuration [67]. Since T cell help is also necessary to initiate somatic hypermutation in B cell through CD40 or CD38 signaling in the germinal center [68], it follows that one important implication of our study is that defective T-B pairing may negatively influence the normal process of germinal center maturation of the B cell response in response to SARS-CoV-2 infection in a critical way.

Which antigens can generate T cell responses depends on the binding specificities of MHC-II molecules, which are highly polymorphic in the human population. We noted a general trend for MHC-II alleles to less effectively present peptides from the RBM region, but also observed some variability across MHC-II supertypes. The main DP and DP2 haplotypes were both common and had the highest potential to present peptides, suggesting that most individuals should carry at least one allele capable of presenting peptides in this region. Which of the two DP haplotypes was more common varied by ancestral population, thus it is possible that differences in the haplotypes could translate to differences in T-B cooperativity levels within groups, though binding affinities for epitopes near the FNCY patch were similar for both. DQ and DR supertypes were less able to present peptides near FNCY, with the exception of DR4, which is among the less common supertypes. Importantly, our analysis was limited to predicted affinity of peptides to MHC-II, and other characteristics such as expression levels, stability or differences in interactions with molecular chaperones likely also contribute to whether FNCY proximal peptides are available to support T-B cooperation [69].

The present study assesses the probability of SARS-CoV-2 peptides of the Spike protein to bind and be presented by MHC-II molecules. Our study is limited by the following: results are an estimate based on an algorithm that encompasses many biophysical variables for MHC-II presentation but certainly not all. In addition, while we believe the epitope containing the FNCY patch is promising for inducing a protective neutralizing response, it is not the sole determinant of a protective antibody response to SARS-CoV-2; as neutralizing antibodies against other portions of the spike and other non-structural proteins have been reported [41, 42, 70–73].

In light of our findings, it can be predicted that, in general, a specific RBM antibody response may be short-lived and that residual immunity from a primary infection may not be sufficient to prevent reinfection after 6–9 months. Sporadic cases of re-infection have been reported by the media in Hong Kong and Nevada [74]. A third case has been reported in a care-home resident who after the second infection produced only low levels of antibodies [75]. Finally, silent re-infections in young workers in a COVID-19 ward who tested positive for the new coronavirus and became reinfected several months later with no symptoms in either instance have been reported [76]. It is tempting to speculate that waning antibody levels or a poorly developed specific NAb antibody response to SARS-CoV-2 can potentially put people at risk of reinfection. Other factors to consider are a bias in the available B cell repertoire in the population and the extent to which a defective T-B cooperation influences the longevity of terminally differentiated plasma cells in the bone marrow [77].

In summary, we provide evidence that MHC-II constrains the CD4 T cell response for epitopes that are best positioned to facilitate T-B pairing in generating and sustaining a potent neutralizing antibody response against a mutationally constrained RBM B cell epitope.

Furthermore, we show that the immunological history of the individual, not necessarily related to infection by other coronaviruses, may confer immunologic advantage. Finally, these findings may have implications for the quality and persistence of a protective, neutralizing antibody response to RBM induced by current SARS-CoV-2 vaccines.

## Materials and methods

Data and code are available at https://github.com/cartercompbio/SARS_CoV_2_T-B_co-op.

### Affinity analysis

NetMHCIIpan version 4.0 was used to predict peptide-MHC-II affinity [78] for generated 15mers along the SARS-CoV-2 spike protein.

### Spike protein analyses

SARS-CoV-2 spike protein sequence and protein regions were obtained from https://www.uniprot.org/uniprot/P0DTC2. Glycan data were obtained from [57] and true-positive sites were aggregated across 3 replicates. To assess depletion of effective binders near the FNCY patch, we performed a Fisher's exact test for binding (median affinity across common alleles <10) versus proximity (+/- 50 amino acids) to FNCY for positions free of glycans. We excluded positions within 10 amino acids of a glycan using the data obtained from Watanabe *et al.* and added a pseudocount of 1.

The SARS1, MERS1, HCoV-229E, HCoV-NL63, HCoV-OC43, and HCoV-HKU1 spike protein sequences were also downloaded from UniProt (P59594, K9N5Q8, P15423, Q6Q1S2, P36334, Q0ZME7, respectively). Multiple sequence alignment was performed on the EMBL-EBI Clustal Omega web server using default parameters [79].

### Structure analysis

The 6M0J 3D X-ray structure for the protein complex containing the SARS-CoV-2 spike protein RBD (P0DTC2) interaction with ACE2 (Q9BYF1) from [45]. The structure figures were prepared using VMD [80].

### Supertype analysis

Supertypes were obtained from [58]. All alpha/beta combinations spanning any of these types were included, resulting in 279 alleles. US supertype frequencies for alleles in DRB1 and DQB1 were obtained from the Be the Match registry [59], US frequencies for alleles in DPB1 were obtained from [60] as DPB1 was not available from the Be the Match registry. Available allele frequencies within each supertype were summed for Fig 3C.

### Motif analysis

All 13-20mer peptides adhering to the following parameters were downloaded from the IEDB [81]: MHC-II assay, positive only, DRB1*01:01 allele, linear peptides; and any peptides with post-translational modifications or noncanonical amino acids were removed. The remaining 10,117 peptides were input into Gibbs cluster v2.0 [82] using the default MHC-II ligand parameters.

## BLAST analysis

15mers were generated along a sliding window +/-30 amino acids from the FNCY patch start and end (455–518, 0-index) and input into NCBI BLAST [83] using the 'refseq_protein' database and excluding SARS-CoV-2 (taxid:2697049). Identified peptides (S2 Table) were then evaluated for binding affinity and any peptide binding to at least one allele was retained for Fig 5D.

## Supporting information

**S1 Table. SARS-CoV-2 neutralizing antibody residues and references used to generate Fig 1D.**
(XLSX)

**S2 Table. BLAST-identified peptides with affinity, and binding fraction.**
(XLSX)

**S1 Fig. Distribution of position scores along the spike protein using the 25th percentile affinity instead of the median affinity.**
(PDF)

**S2 Fig. Overview of subject peptides that bind at least one retrieved from BLAST search.**
(A) Pileup of corresponding query peptides' start positions of BLAST-identified peptides that bind to at least one common MHC-II allele. The below barplot shows Fig 3A for reference: the aggregated position scores across supertypes for positions proximal to FNCY. (B) Scatterplot showing the median supertype affinities of BLAST-identified peptides that may bind (median affinity <20) along the corresponding start positions of queried peptides along the spike protein. The FNCY motif region is highlighted in grey. (C) Clustermap showing the median supertype affinities of BLAST-identified peptides that may bind (median affinity <20) to at least one supertype. Median affinities greater than 20 have been adjusted to 20 for better visualization of binding peptides.
(PDF)

**S1 Graphical abstract.**
(TIF)

## Acknowledgments

The graphical abstract was created using BioRender and used the PDB [84] structure 6VXX from [85].

## Author Contributions

**Conceptualization:** Maurizio Zanetti, Hannah Carter.

**Data curation:** Andrea Castro, Kivilcim Ozturk.

**Formal analysis:** Andrea Castro, Kivilcim Ozturk.

**Funding acquisition:** Maurizio Zanetti, Hannah Carter.

**Investigation:** Maurizio Zanetti, Hannah Carter.

**Methodology:** Maurizio Zanetti, Hannah Carter.

**Project administration:** Maurizio Zanetti, Hannah Carter.

**Software:** Andrea Castro, Kivilcim Ozturk.

**Supervision:** Maurizio Zanetti, Hannah Carter.

**Validation:** Andrea Castro, Kivilcim Ozturk.

**Visualization:** Andrea Castro, Kivilcim Ozturk.

**Writing – original draft:** Andrea Castro, Kivilcim Ozturk, Maurizio Zanetti, Hannah Carter.

**Writing – review & editing:** Andrea Castro, Maurizio Zanetti, Hannah Carter.

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
