## [Decision Letter · Decision Letter 0]

30 Dec 2020

PONE-D-20-38239

MHC-II constrains the natural neutralizing antibody response to the SARS-CoV-2 spike RBM in humans

PLOS ONE

Dear Dr. Zanetti,

Thank you for submitting your manuscript to PLOS ONE. After careful consideration, we feel that it has merit but does not fully meet PLOS ONE’s publication criteria as it currently stands. Therefore, we invite you to submit a revised version of the manuscript that addresses the points raised during the review process.

Please pay particular attention to the comments made by both the reviewers on the lack of experimental evidence to support the predictive data presented in the manuscript. Please discuss this appropriately in the revised version in the line with the comments made by the reviewers.

We look forward to receiving your revised manuscript.

Kind regards,

Jayanta Bhattacharya

Academic Editor

PLOS ONE

Journal Requirements:

2. To comply with PLOS ONE submission guidelines, in your Methods section, please provide additional information regarding your statistical analyses.

For more information on PLOS ONE's expectations for statistical reporting, please see https://journals.plos.org/plosone/s/submission-guidelines.#loc-statistical-reporting

3. Please ensure you have discussed any potential limitations of your study in the Discussion.

Additional Editor Comments:

Please pay particular attention to both the reviewers comments on the lack of experimental evidence to support the predictive data presented in this manuscript. Please discuss this when you revise your manuscript in the line with the reviewers comments.

Reviewers' comments:

Reviewer's Responses to Questions

**Comments to the Author**

1. Is the manuscript technically sound, and do the data support the conclusions?

Reviewer #1: Yes

Reviewer #2: Partly

2. Has the statistical analysis been performed appropriately and rigorously? 

Reviewer #1: Yes

Reviewer #2: Yes

3. Have the authors made all data underlying the findings in their manuscript fully available?

Reviewer #1: Yes

Reviewer #2: Yes

4. Is the manuscript presented in an intelligible fashion and written in standard English?

Reviewer #1: Yes

Reviewer #2: Yes

5. Review Comments to the Author

Reviewer #1: Andrea Castro et al, have performed a prediction analysis of the binding efficiency of peptides (proximal to the RBM), identified to be potential B cell epitopes targeted by neutralizing antibodies, to MHC-II alleles and their key observation was a poor binding interaction and a limited availability of effective T cell epitopes in close proximity to the RBM B cell epitopes. Based on their findings, the authors suggest that this lack of MHC-II binding may impact the B cell mediated antigen specific and MHC restricted T cell activation, and limit the CD4 T cell help that is required for affinity maturation of the B cells in the germinal centres, plausibly leading to less potent neutralizing antibodies and limited memory responses.

An extensive ex vivo functional analysis by Jose Mateus, also referred to herein, suggested that previous exposures to other human coronaviruses could potentially generate protective immunity toward SARS-CoV-2 as they found relatively few responses of pre-existing CD4 T cell responses to peptides proximal to the FNCY patch in the RBM, derived from the spike protein in the T cells from unexposed individuals. The authors in this study further identified peptides associated with multiple microbial organisms that plausibly meet the criteria to potentially generate CD4 T cell memory relevant to the RBM of SARS-CoV-2. Based on these observations, the authors conclude that MHC II constrains the CD4 responses towards neutralizing determinants that are in close proximity to facilitate interaction between antigen specific T and B cell, required for the generation of potent neutralizing antibodies and memory cells. Further, the above factors may be responsible for the short lived RBM directed neutralizing antibody responses and that memory response of an individual, not necessarily related to infection by other coronaviruses, may confer immunologic advantage from a primary infection.

The findings of this study are interesting and add to the growing body of information on factors that plausibly influence the pattern of humoral immune responses and limited memory observed during natural infection. However, the information generated in this study is based on prediction analysis and not backed by ex vivo experiments to demonstrate the limited ability of the peptides in the RBM motif to activate T cells due to poor binding to MHCII. This should be discussed as a limitation of the study and the title should be modified accordingly.

Reviewer #2: The manuscript presents an interesting hypothesis and accompanying exclusive in silico predictions in support. These, in my opinion, are not enough to warrant publication without the addition of at least some validation data (retrospective meta analysis of COVID-19 data and other viral infections; immunogenicity experiments) as detailed in the comments below.

The introduction does not clearly delineate the influence of T-B cooperation on 1) memory cell generation and persistence and 2) potent neutralizing responses as two possibly mutually exclusive events. Both are dependent on T-B cooperation but not necessary linked. This confuses the rationale of the study. Authors should consider clearly state what aspect their study attempts to shed light on. Are they trying to say that proximal epitope/neutralization paratope combinations are elicitors of protective responses restricted by MHCII?

Lines 98-103: This is not necessarily supportive of the link between memory cell generation/persistence (‘high magnitude’) and neutralizing responses. Also, does the occurrence of the ‘minority’ proximal MHCII peptide specific CD4+T cell population correlate with the aforementioned aspects of humoral immunity in infected/convalescent individuals? An analysis of this sort could strengthen the weight of the conclusions in this manuscript.

Why do the authors assume a single neturalizing epitope, the one they focus on, is the sole determinant of a protective response? While their exclusively in silico results generally (but not overwhelmingly) support the hypothesis, this study should have included some in vivo data (even basic immunization studies in small animals) to validate the predicted defect in presentation of proximal (to the neutralization paratope) epitopes. Also, comparative (retrospective??) analysis of other viral infections where much more is known about T-B cooperation would strengthen the claims of the manuscript. These are too generalized in implication not to have supporting in vivo data considering the large gaps and mainly empirical knowledge that exists regarding the generation of protective anti-viral responses.

6. PLOS authors have the option to publish the peer review history of their article (what does this mean?). If published, this will include your full peer review and any attached files.

Reviewer #1: No

Reviewer #2: No

---

## [Author Response · Author response to Decision Letter 0]

23 Jan 2021

Reviewer #1: Andrea Castro et al, have performed a prediction analysis of the binding efficiency of peptides (proximal to the RBM), identified to be potential B cell epitopes targeted by neutralizing antibodies, to MHC-II alleles and their key observation was a poor binding interaction and a limited availability of effective T cell epitopes in close proximity to the RBM B cell epitopes. Based on their findings, the authors suggest that this lack of MHC-II binding may impact the B cell mediated antigen specific and MHC restricted T cell activation, and limit the CD4 T cell help that is required for affinity maturation of the B cells in the germinal centres, plausibly leading to less potent neutralizing antibodies and limited memory responses.

An extensive ex vivo functional analysis by Jose Mateus, also referred to herein, suggested that previous exposures to other human coronaviruses could potentially generate protective immunity toward SARS-CoV-2 as they found relatively few responses of pre-existing CD4 T cell responses to peptides proximal to the FNCY patch in the RBM, derived from the spike protein in the T cells from unexposed individuals. The authors in this study further identified peptides associated with multiple microbial organisms that plausibly meet the criteria to potentially generate CD4 T cell memory relevant to the RBM of SARS-CoV-2. Based on these observations, the authors conclude that MHC II constrains the CD4 responses towards neutralizing determinants that are in close proximity to facilitate interaction between antigen specific T and B cell, required for the generation of potent neutralizing antibodies and memory cells. Further, the above factors may be responsible for the short lived RBM directed neutralizing antibody responses and that memory response of an individual, not necessarily related to infection by other coronaviruses, may confer immunologic advantage from a primary infection.

The findings of this study are interesting and add to the growing body of information on factors that plausibly influence the pattern of humoral immune responses and limited memory observed during natural infection. However, the information generated in this study is based on prediction analysis and not backed by ex vivo experiments to demonstrate the limited ability of the peptides in the RBM motif to activate T cells due to poor binding to MHCII. This should be discussed as a limitation of the study and the title should be modified accordingly. 

We thank the reviewer for this feedback and have sought to emphasize the computational nature of the analysis in the discussion as well as the title. The following has been added to the discussion:

● “The present study assesses the probability of SARS-CoV-2 peptides of the Spike protein to bind and be presented by MHC-II molecules. Our study is limited by the following: results are an estimate based on an algorithm that encompasses many biophysical variables for MHC-II presentation but certainly not all. In addition, while we believe the epitope containing the FNCY patch is promising for inducing a protective neutralizing response, it is not the sole determinant of a protective antibody response to SARS-CoV-2; as neutralizing antibodies against other portions of the spike and other non-structural proteins have been reported (Yuan et al. 2020; Pinto et al. 2020; Wang et al. 2020; McAndrews et al. 2020; Okba et al. 2020; Fenwick et al. 2021).”

We have also revised the title to:

● In silico analysis suggests less effective MHC-II presentation of SARS-CoV-2 RBM peptides: Implication for neutralizing antibody responses

 

Reviewer #2: The manuscript presents an interesting hypothesis and accompanying exclusive in silico predictions in support. These, in my opinion, are not enough to warrant publication without the addition of at least some validation data (retrospective meta analysis of COVID-19 data and other viral infections; immunogenicity experiments) as detailed in the comments below.

The introduction does not clearly delineate the influence of T-B cooperation on 1) memory cell generation and persistence and 2) potent neutralizing responses as two possibly mutually exclusive events. Both are dependent on T-B cooperation but not necessary linked. This confuses the rationale of the study. Authors should consider clearly state what aspect their study attempts to shed light on. Are they trying to say that proximal epitope/neutralization paratope combinations are elicitors of protective responses restricted by MHCII?

We thank the reviewer for this comment. We have revised the introduction to clarify that memory cell generation does not necessarily imply a potent neutralizing response, but that an optimal neutralizing response should entail memory cell generation and persistence. We have clarified our hypothesis: because memory B cell maturation heavily depends on interaction with a CD4 T cell, peptides proximal to or including the B cell epitope need to be effectively presented via MHC-II. Furthermore, this process needs to happen for B cells that produce neutralizing antibodies targeting the RBM; and their differentiation into memory B cells. Memory responses for non-neutralizing antibodies may occur but are not pivotal to protection. On the other hand, it is still poorly understood whether early neutralizing antibodies in COVID-19 patients lead to memory B cells producing antibodies of the same specificity. Therefore one could argue that the inability to present sequences near putative B cell epitopes bound by neutralizing antibodies may inhibit memory cell generation and affect the strength of neutralizing antibody response. 

The rationale is clarified in the introduction as follows:

● “Specifically, we hypothesize that the inability to present SARS-CoV-2 peptide sequences near putative B cell epitopes may impair memory cell generation and consequently reduce the strength and longevity of overall and neutralizing antibody responses.”

Initial COVID-19 responses appear to rely largely on early activated B cells that produce antibodies in quasi-germline configuration and use a restricted VH rearrangement (IGHV3-23 and IGHV3-7), suggesting that these cells are ‘innate-like B cells’ (Wen et al. 2020; Ju et al. 2020; Liu et al. 2020; Tortorici et al. 2020) and have not undergone somatic hypermutation and maturation. Little is known if these early activated B cells expand in patients or if they fully mature, maintaining antibody specificity. We have added this to the introduction:

● “Early activated B cells produce antibodies in quasi-germline configuration and are likely ‘innate-like B cells’ (Wen et al. 2020; Ju et al. 2020; Liu et al. 2020; Tortorici et al. 2020) that have not undergone somatic hypermutation and maturation.”

Germline/innate B cells might provide protection initially but will decay with time, thus memory B cells are required for long term protection, a process that depends on T-B cooperation at germinal centers. A recent study has shown there is a lack of germinal center type B cells in COVID-19 patients, which will affect long-lived memory and high-affinity B cells (Kaneko et al. 2020). We have added this new reference to the introduction:

● “Consistent with the above argument, a lack of germinal center formation but robust activation of non-germinal type B cells has been reported in cases of severe COVID-19 infection, impairing production of long-lived memory or high affinity B cells (Kaneko et al. 2020).”

Lines 98-103: This is not necessarily supportive of the link between memory cell generation/persistence (‘high magnitude’) and neutralizing responses. Also, does the occurrence of the ‘minority’ proximal MHCII peptide specific CD4+T cell population correlate with the aforementioned aspects of humoral immunity in infected/convalescent individuals? An analysis of this sort could strengthen the weight of the conclusions in this manuscript.

We have reviewed lines 98-103 and have updated the text for clarity. We meant for this section to emphasize the finding in (Mateus et al. 2020) that CD4 T cell responses by T cells obtained from unexposed individuals largely occur outside of the RBD region (Figure 4). Separately, an earlier study by Ni et al. has observed that numbers of IFN-γ-secreting RBD-specific T cells were much lower than those of nucleocapsid protein (NP) specific T cells (Ni et al. 2020). 

Updated text:

● “Antibody responses against SARS-CoV-2 depend on CD4 T cell help. Spike-specific CD4 T cell responses have been found to correlate with the magnitude of the anti-RBD IgG response whereas non-spike CD4 T cell responses do not (44). However, in unexposed patients, spike-specific CD4 T cells reactive with MHC-II peptides proximal to the central B cell epitope represent a minority (~10%) of the total CD4 T cell responses, which are dominated by responses against either the distal portion of the spike protein or other structural antigens (45). Surprisingly, these CD4 T cell responses are largely cross-reactive and originate from previous coronavirus infections (46).”

Why do the authors assume a single neturalizing epitope, the one they focus on, is the sole determinant of a protective response? While their exclusively in silico results generally (but not overwhelmingly) support the hypothesis, this study should have included some in vivo data (even basic immunization studies in small animals) to validate the predicted defect in presentation of proximal (to the neutralization paratope) epitopes. 

We thank the reviewer for this question. We have revised the language in the introduction and added to the discussion (below, respectively) to clarify that this epitope is promising for a protective response, but is not the sole determinant of one. 

● “...we decided to test the hypothesis that associative recognition of a key RBM B cell epitope (in and around the FNCY patch) and proximal MHC-II-restricted epitopes may be defective with detrimental effects on preferential T-B pairing.” 

● “In addition, while we believe the epitope containing the FNCY patch is promising for inducing a protective neutralizing response, it is not the sole determinant of a protective antibody response to SARS-CoV-2; as neutralizing antibodies against other portions of the spike and other non-structural proteins have been reported (Yuan et al. 2020; Pinto et al. 2020; Wang et al. 2020; McAndrews et al. 2020; Okba et al. 2020; Fenwick et al. 2021).”

We focus on the FNCY patch for a few reasons: (1) in a competition binding assay with other neutralizing antibodies (NAbs), NAbs that came in contact with this region outperformed NAbs that did not across samples from hospitalized, symptomatic, and asymptomatic patients (Figure R1 taken from (Piccoli et al. 2020)). (2) Monoclonal antibodies that have been generated by COVID-19 patients tend to neutralize by binding to the region involving the FNCY patch as shown in Figure 1D in the manuscript. Furthermore, it was recently shown that residue E484, which is 2aa upstream of the FNCY patch, accounts for most of the neutralization via polyclonal serum antibodies targeting the RBD (Greaney et al. 2021), highlighting the importance of this region. This does not exclude the possibility that antibodies against other portions of the spike including those generated against previous coronaviruses may also neutralize (Yuan et al. 2020). 

We have added additional text in the introduction to clarify our focus on the FNCY patch:

● “NAbs that make contact with the FNCY patch outperform other NAbs that do not in competition binding assays, highlighting the importance of the region in neutralizing ACE2 binding (Piccoli et al. 2020)”

Figure R1. (Figure 7H from (Piccoli et al. 2020)). S2H14 and S2H13 antibodies span the FNCY patch (highlighted in the red box) while the remaining antibodies do not. This figure describes the results of a competition binding assay to ACE2 for hospitalized, symptomatic, and asymptomatic patients’ sera. 

We agree with the reviewer that validating our prediction in vivo is ideal. While we do not have access to appropriate humanized transgenic mice, we found a recent study where Prakash et al. immunized HLA-A02/HLA-DRB1 double transgenic mice and found that the B cell epitope S471-501 (spanning the FNCY patch) induced a low number of Antibody Secreting Cells (ASC) compared to other spike or non-spike peptides (Figure R2). This epitope was also variably recognized by antibodies in sera from HLA-A02 positive SARS-CoV-2 infected individuals, though comprehensive MHC-I and MHC-II data was not available for the human samples and likely affected their results (Figure R2). These results, albeit limited to HLA-DRB1*0101 in mice and unknown MHC-II types in the human samples, are consistent with our prediction that it is harder to get memory B cell responses to this epitope. Importantly, we believe that the conclusions made in the manuscript based on a global in silico analysis of the MHC-II alleles would not be possible to validate in the mouse because there are very few HLA Class II transgenic mice available that lack endogenous HLA, and any result would be by definition of very limited value compared to the breadth of our study. 

Figure R2. (Figure 9 from (Prakash et al. 2020)). (Top left) barplot of antibody secreting cell (ASC) levels for various B cell epitopes along the spike protein in HLA-A02/DRB1 mice. (Top right) barplot of IgG response levels for various B cell epitopes along the spike protein in HLA-A02/DRB1 mice. (Bottom) barplot of IgG response levels for COVID-19 patient sera against various B cell epitopes along the spike protein. Red arrows indicate the S471-501 epitope that spans the FNCY patch.

Also, comparative (retrospective??) analysis of other viral infections where much more is known about T-B cooperation would strengthen the claims of the manuscript. These are too generalized in implication not to have supporting in vivo data considering the large gaps and mainly empirical knowledge that exists regarding the generation of protective anti-viral responses.

We thank the reviewer for this comment. We now include more references to studies on a variety of viruses in which T-B cooperation has been studied in the mouse: 

● In the influenza A virus (PR8) system it was shown that while Th1 CD4 T cell responses on their own are ineffective at promoting recovery from infection, antibodies generated through T-B cooperation were indispensable in the protective response against the virus (Mozdzanowska et al. 1997). Subsequent studies using a different influenza A strain confirmed the relevance of T-B cooperation and that CD4 T cells represent a limiting factor in the kinetics and early magnitude of the primary B cell response to virus challenge and provide help in a preferential way (i.e. intra-molecular but nor inter-molecular) (Alam et al. 2014). 

● Additional information comes from studies examining the role CD40-CD40L interaction in T-B cooperation. The CD40-CD40L interaction is required for the generation of antibody responses, isotype switching and generation of memory responses, to T-dependent antigens in non-viral model systems (Parker 1993). In a two virus model, LCMV (lymphocytic choriomeningitis virus) and VSV (vesicular stomatitis virus) T-B cooperation was shown to be necessary for antiviral protection and require CD40-CD40L interactions (Oxenius et al. 1996). Interestingly, this study also showed that the activation of CD4 T cells not associated with the activation of B cells (e.g., inflammatory reaction) and antibody production was not compromised (Oxenius et al. 1996).

We have added the following text to the introduction:

● “The relevance of T-B cooperation in protective antiviral responses has been documented in numerous systems. In the influenza A virus (PR8) system it was shown that while Th1 CD4 T cell responses on their own are ineffective at promoting recovery from infection, antibodies generated through T-B cooperation were indispensable in the protective response against the virus (Mozdzanowska et al. 1997). In a different influenza A strain, it was shown that T-B cooperation and CD4 T cells represent a limiting factor in the kinetics and early magnitude of the primary B cell response to virus challenge and provide help in a preferential way (i.e. intra-molecular but nor inter-molecular) (Alam et al. 2014). Additionally, CD40-CD40L (costimulatory molecules found on B cells and CD4 T cells, respectively) interaction is required for the generation of antibody responses, isotype switching and memory responses in non-viral model systems (Parker 1993). In LCMV (lymphocytic choriomeningitis virus) and VSV (vesicular stomatitis virus) abrogation of CD40-CD40L interaction prevented T-B cooperation and thus inhibited antiviral protection (Oxenius et al. 1996). Interestingly, this study also showed that the activation of CD4 T cells (e.g., inflammatory CD4 T cells) not associated with the activation of B cells was not compromised (Oxenius et al. 1996). These data demonstrate the relevance of T-B cooperation in the antibody response in protection against viral infection.”

In addition, we performed retrospective analysis on 100 COVID-19 bulk RNA-seq samples from (Overmyer et al. 2020). We found that patients who received mechanical ventilation or who were treated in the ICU had lower levels of various co-stimulatory molecules or cytokines indicative of T-B cooperation (Figure R3). This suggests impairment of CD40-mediated class switching (ICOS), T-cell dependent B cell proliferation (OX40) and T cell activation and proliferation (CD28) as well as lack of T cell activation (IL4) contribute to poorer outcomes.

Figure R3. Gene expression of co-stimulatory molecules CD40, OX40, CD28, ICOS and cytokine IL4 for COVID-19 patients (top panel) treated in or out of the ICU or (bottom panel) who received mechanical ventilation.

Alam, Shabnam, Zackery A. G. Knowlden, Mark Y. Sangster, and Andrea J. Sant. 2014. “CD4 T Cell Help Is Limiting and Selective during the Primary B Cell Response to Influenza Virus Infection.” Journal of Virology 88 (1): 314–24.

Greaney, Allison J., Andrea N. Loes, Katharine H. D. Crawford, Tyler N. Starr, Keara D. Malone, Helen Y. Chu, and Jesse D. Bloom. 2021. “Comprehensive Mapping of Mutations to the SARS-CoV-2 Receptor-Binding Domain That Affect Recognition by Polyclonal Human Serum Antibodies.” Cold Spring Harbor Laboratory. https://doi.org/10.1101/2020.12.31.425021.

Ju, Bin, Qi Zhang, Jiwan Ge, Ruoke Wang, Jing Sun, Xiangyang Ge, Jiazhen Yu, et al. 2020. “Human Neutralizing Antibodies Elicited by SARS-CoV-2 Infection.” Nature 584 (7819): 115–19.

Kaneko, Naoki, Hsiao-Hsuan Kuo, Julie Boucau, Jocelyn R. Farmer, Hugues Allard-Chamard, Vinay S. Mahajan, Alicja Piechocka-Trocha, et al. 2020. “Loss of Bcl-6-Expressing T Follicular Helper Cells and Germinal Centers in COVID-19.” Cell 183 (1): 143–57.e13.

Liu, Lihong, Pengfei Wang, Manoj S. Nair, Jian Yu, Micah Rapp, Qian Wang, Yang Luo, et al. 2020. “Potent Neutralizing Antibodies against Multiple Epitopes on SARS-CoV-2 Spike.” Nature 584 (7821): 450–56.

Mateus, Jose, Alba Grifoni, Alison Tarke, John Sidney, Sydney I. Ramirez, Jennifer M. Dan, Zoe C. Burger, et al. 2020. “Selective and Cross-Reactive SARS-CoV-2 T Cell Epitopes in Unexposed Humans.” Science, August. https://doi.org/10.1126/science.abd3871.

Mozdzanowska, K., M. Furchner, K. Maiese, and W. Gerhard. 1997. “CD4+ T Cells Are Ineffective in Clearing a Pulmonary Infection with Influenza Type A Virus in the Absence of B Cells.” Virology 239 (1): 217–25.

Ni, Ling, Fang Ye, Meng-Li Cheng, Yu Feng, Yong-Qiang Deng, Hui Zhao, Peng Wei, et al. 2020. “Detection of SARS-CoV-2-Specific Humoral and Cellular Immunity in COVID-19 Convalescent Individuals.” Immunity 52 (6): 971–77.e3.

Overmyer, Katherine A., Evgenia Shishkova, Ian J. Miller, Joseph Balnis, Matthew N. Bernstein, Trenton M. Peters-Clarke, Jesse G. Meyer, et al. 2020. “Large-Scale Multi-Omic Analysis of COVID-19 Severity.” Cell Systems, October. https://doi.org/10.1016/j.cels.2020.10.003.

Oxenius, A., K. A. Campbell, C. R. Maliszewski, T. Kishimoto, H. Kikutani, H. Hengartner, R. M. Zinkernagel, and M. F. Bachmann. 1996. “CD40-CD40 Ligand Interactions Are Critical in T-B Cooperation but Not for Other Anti-Viral CD4+ T Cell Functions.” The Journal of Experimental Medicine 183 (5): 2209–18.

Parker, D. C. 1993. “The Functions of Antigen Recognition in T Cell-Dependent B Cell Activation.” Seminars in Immunology 5 (6): 413–20.

Piccoli, Luca, Young-Jun Park, M. Alejandra Tortorici, Nadine Czudnochowski, Alexandra C. Walls, Martina Beltramello, Chiara Silacci-Fregni, et al. 2020. “Mapping Neutralizing and Immunodominant Sites on the SARS-CoV-2 Spike Receptor-Binding Domain by Structure-Guided High-Resolution Serology.” Cell, September. https://doi.org/10.1016/j.cell.2020.09.037.

Prakash, Swayam, Ruchi Srivastava, Pierre-Gregoire Coulon, Nisha R. Dhanushkodi, Aziz A. Chentoufi, Delia F. Tifrea, Robert A. Edwards, et al. 2020. “Genome-Wide Asymptomatic B-Cell, CD4+ and CD8+ T-Cell Epitopes, That Are Highly Conserved Between Human and Animal Coronaviruses, Identified from SARS-CoV-2 as Immune Targets for Pre-Emptive Pan-Coronavirus Vaccines.” Cold Spring Harbor Laboratory. https://doi.org/10.1101/2020.09.27.316018.

Tortorici, M. Alejandra, Martina Beltramello, Florian A. Lempp, Dora Pinto, Ha V. Dang, Laura E. Rosen, Matthew McCallum, et al. 2020. “Ultrapotent Human Antibodies Protect against SARS-CoV-2 Challenge via Multiple Mechanisms.” Science, September. https://doi.org/10.1126/science.abe3354.

Wen, Wen, Wenru Su, Hao Tang, Wenqing Le, Xiaopeng Zhang, Yingfeng Zheng, Xiuxing Liu, et al. 2020. “Immune Cell Profiling of COVID-19 Patients in the Recovery Stage by Single-Cell Sequencing.” Cell Discovery 6 (May): 31.

Yuan, Meng, Nicholas C. Wu, Xueyong Zhu, Chang-Chun D. Lee, Ray T. Y. So, Huibin Lv, Chris K. P. Mok, and Ian A. Wilson. 2020. “A Highly Conserved Cryptic Epitope in the Receptor Binding Domains of SARS-CoV-2 and SARS-CoV.” Science 368 (6491): 630–33.

---

## [Editor Report · Decision Letter 1]

26 Jan 2021

In silico analysis suggests less effective MHC-II presentation of SARS-CoV-2 RBM peptides: implication for neutralizing antibody responses

PONE-D-20-38239R1

Dear Dr. Zanetti,

We’re pleased to inform you that your manuscript has been judged scientifically suitable for publication and will be formally accepted for publication once it meets all outstanding technical requirements.

Kind regards,

Jayanta Bhattacharya

Academic Editor

PLOS ONE
---

## [Editor Report · Acceptance letter]

2 Feb 2021

PONE-D-20-38239R1 

In silico analysis suggests less effective MHC-II presentation of SARS-CoV-2 RBM peptides: implication for neutralizing antibody responses  

Dear Dr. Zanetti:

I'm pleased to inform you that your manuscript has been deemed suitable for publication in PLOS ONE. Congratulations! Your manuscript is now with our production department. 

Kind regards, 

on behalf of

Dr. Jayanta Bhattacharya 

Academic Editor

PLOS ONE